

# Characterization of the artisanal fishing communities in Nepal and potential implications for the conservation and management of Ganges River Dolphin (*Platanista gangetica gangetica*)

Shambhu Paudel[1], Juan C. Levesque[2], Camilo Saavedra[3], Cristina Pita[4] and Prabhat Pal[5]

[1] Kathmandu Forestry College and The Himalaya, Kathmandu, Nepal
[2] Environmental Resources Management, Impact, Assessment, and Planning Division, Tampa, Florida, United States
[3] Instituto Español de Oceanografía, CO Vigo, Spain, Spain
[4] Department of Environment and Planning & Center for Environmental and Marine Studies (CESAM), University of Aveiro, Aveiro, Portugal
[5] Department of Soil Conservation and Watershed Management, Ministry of Forest and Soil Conservation, Nepal, The Himalaya, Kathmandu, Nepal

Corresponding author
Shambhu Paudel,
oasis.excurrent@gmail.com

## ABSTRACT

The Ganges River dolphin (*Platanista gangetica gangetica*) (GRD) is classified as one of the most endangered of all cetaceans in the world and the second scarcest freshwater cetacean. The population is estimated to be less than 2,000 individuals. In Nepal's Narayani, Sapta Koshi, and Karnali river systems, survival of GRD continues to be threatened by various anthropogenic activities, such as dam construction and interactions with artisanal fisheries. A basic description of the geographic scope, economics, and types of gear used in these fisheries would help managers understand the fishery-dolphin interaction conflict and assist with developing potential solutions. The main goal was to provide new information on the artisanal fishing communities in Nepal. The specific objectives were to identify, compile, and investigate the demographics, economics, fishing characteristics, and perception of fishermen about GRD conservation in the Narayani, Sapta Koshi, and Karnali rivers so conservation managers can develop and implement a potential solution to the GRD-fishery interaction problem in Nepal. Based on 169 interviews, 79% of Nepalese fishermen indicated fishing was their primary form of income. Fishermen reported fishing effort was greater in summer than winter; greatest in the afternoon (14:30 hrs ± 0:27) and during low water level conditions; and gear was set 4.8 ± 0.2 days/week. Fishermen reported using eight different types of monofilament nets (gillnets and cast nets). Sixty percent used gillnets less than 10 m long, and nearly 30% preferred gillnets between 10 and 100 m long; a few used gillnets longer than 100 m. Most fishermen reported they believed education, awareness, and changing occupations were important for GRD conservation, but they indicated that alternative occupational options were currently limited in Nepal. Nepalese fishermen acknowledged that fisheries posed a risk to GRD, but they believed water pollution, and dam/irrigation developments were the greatest threats.

## INTRODUCTION

The Ganges River dolphin (*Platanista gangetica gangetica*) (GRD) is classified as one of the most endangered of all cetaceans in the world and the second scarcest freshwater cetacean (*Reeves, Smith & Kasuya, 2000*; *Sinha, Behera & Choudhary, 2010*; *IUCN, 2012*). According to *Smith & Braulik (2012)*, the population is estimated to be less than 2,000 individuals. Similar to other cetaceans, the GRD is long-lived (~30 years), matures late, and gives birth to a limited number of calves (1–2 per calving) (*IUCN, 2012*). At one time, this freshwater cetacean was primarily found in the Ganges and Brahmaputra rivers, including several associated tributaries in Bangladesh, India, and Nepal (*Jones, 1982*). Today, the Ganges river has the largest remaining population (*Smith, 1993*). In Nepal, the remaining viable population is restricted to the Karnali, Narayani and Sapta Koshi river systems (*Smith et al., 1994*; *Timilsina, Tamang & Baral, 2003*; *WWF Nepal Program, 2006*; *Paudel et al., 2015*).

The GRD is vulnerable to various anthropogenic activities because they are usually found in some of the most densely populated regions (*Smith & Braulik, 2012*); the population of Nepal is 27.8 million. Nepalese river-dependent communities continue to grow and expand, so it is no surprise that most of the GRD-human interaction issues are associated with these areas (*CBS, 2003*), which escalates the human-dolphin interface dilemma. Based on *Paudel (2012)*, the main threat to GRD is probably habitat fragmentation caused by the construction of dams, but it is likely that other human-induced activities (e.g., fishing, pollution and habitat loss) have also led to the decline of the GRD population. Besides the construction of dams, the lack of river and watershed management (open-access resource exploitation) and the geographical expansion of artisanal fisheries are the greatest threats to GRD (*Dudgeon, 2000*; *Manel, Buckton & Ormerod, 2000*; *Gergel et al., 2002*). Because most Nepalese are completely dependent on natural resources for income and survival, some basic daily activities threaten the conservation and recovery of the GRD, such as artisanal fisheries (*Berkes, 1985*; *Turvey et al., 2007*). Unfortunately, the GRD continues to be directly targeted by some fishermen for its oil and meat; the oil is used as bait in a few fisheries and the meat is consumed (*Sinha, Behera & Choudhary, 2010*). The species is also incidentally injured or killed in gillnets (*Reeves, Leatherwood & Mohan, 1993*; *Smith, 1993*). In 2013, a GRD was found dead in the Karnali river (Lalmati area) that was later linked to gillnet gear (*Paudel et al., 2015*). Another threat to the GRD in Nepal is direct competition with fishermen. *Kelkar et al. (2010)* reported that fishermen compete with GRD because they target various species of fish that are essential to the GRD's diet, such as mullet (*Rhinomugil corsula*) or siloroid catfish (*Bagarius bagarius*) (*Smith, 1993*).

A conservation action plan was developed and implemented in India to conserve, protect, and recover the GRD (*Sinha, Behera & Choudhary, 2010*); however, the species has received limited management attention in other regions, such as

Nepal (*Jnawali et al., 2011*). Recently, the Nepalese government began re-enforcing the mandates of the Department of National Parks and Wildlife Conservation Act of 1973 and designated several protected areas in the Karnali (Bardiya National Park), Sapta Koshi (Koshi Tappu Wildlife Reserve), and Narayani (Chitwan National Park) river systems to protect the species. Despite implementing these conservation measures, the GRD population continues to decline at an alarming rate in Nepal (*Jnawali & Bhuju, 2000*). Officials understand that artisanal fisheries are an issue for the conservation and recovery of the GRD, but fishery management or strategies for reducing GRD-fishery interactions are currently lacking. Basic information describing artisanal fisheries and activity is essential for understanding the GRD-fishery problem and developing a potential solution (*Rojas-Bracho & Reeves, 2013*). Regrettably, this type of information is usually unavailable and challenging to obtain, especially in developing countries, such as Nepal. Given the lack of information, the main goal of the present research was to provide new information on the artisanal fishing communities in Nepal. The specific objectives were to identify, compile, and investigate the demographics, economics, fishing characteristics, and perception of fishermen about GRD conservation in the Narayani, Sapta Koshi, and Karnali rivers so conservation managers can develop and implement a potential solution to the GRD-fishery interaction problem in Nepal.

## MATERIALS AND METHODS

### Study area

The survey was conducted in four districts (Bardiya, Nabalparasi, Saptari and Sunsari) situated along three main rivers (Narayani, Sapta Koshi and Karnali) in Nepal. The Bardiya, Nabalparasi, Saptari, and Sunsari districts within our study area represented 45 villages located 1 km of the riverbank (Fig. 1). We chose this region to survey because these river systems serve as habitat for the GRD in Nepal. In addition, these three rivers are major tributaries of the Ganges River. All of these rivers are located downstream of the Siwalik foothills of the Nepalese Himalayas, which represents the upstream limit of GRD distribution in southern Asia. Seasonal snow melt in the Himalayas controls much of the fluctuating water levels in these rivers. Fluctuations in water level cause dolphins to migrate downstream through the barrages during flood periods. For the purpose of this study, we defined various sections of the river as following: (1) the main channel mid area was the center of the main river or tributary; this region of the river has the fastest water velocity; (2) the main channel near the riverbank was the location where the water velocity and depth were lower than the center of the river; and (3) the area behind sandbars/islands was as a parcel of land with sandbars surrounded by water on all sides. The confluence area was located downstream and the distributary area was located upstream.

### Survey methods

Fishery and socio-economic information was collected using a face-to-face questionnaire approach with registered (fishing associations) fishermen located along the Narayani, Sapta Koshi, and Karnali rivers in Nepal during August 2013. We specifically chose to interview registered fishermen because fishermen associations represented a large number
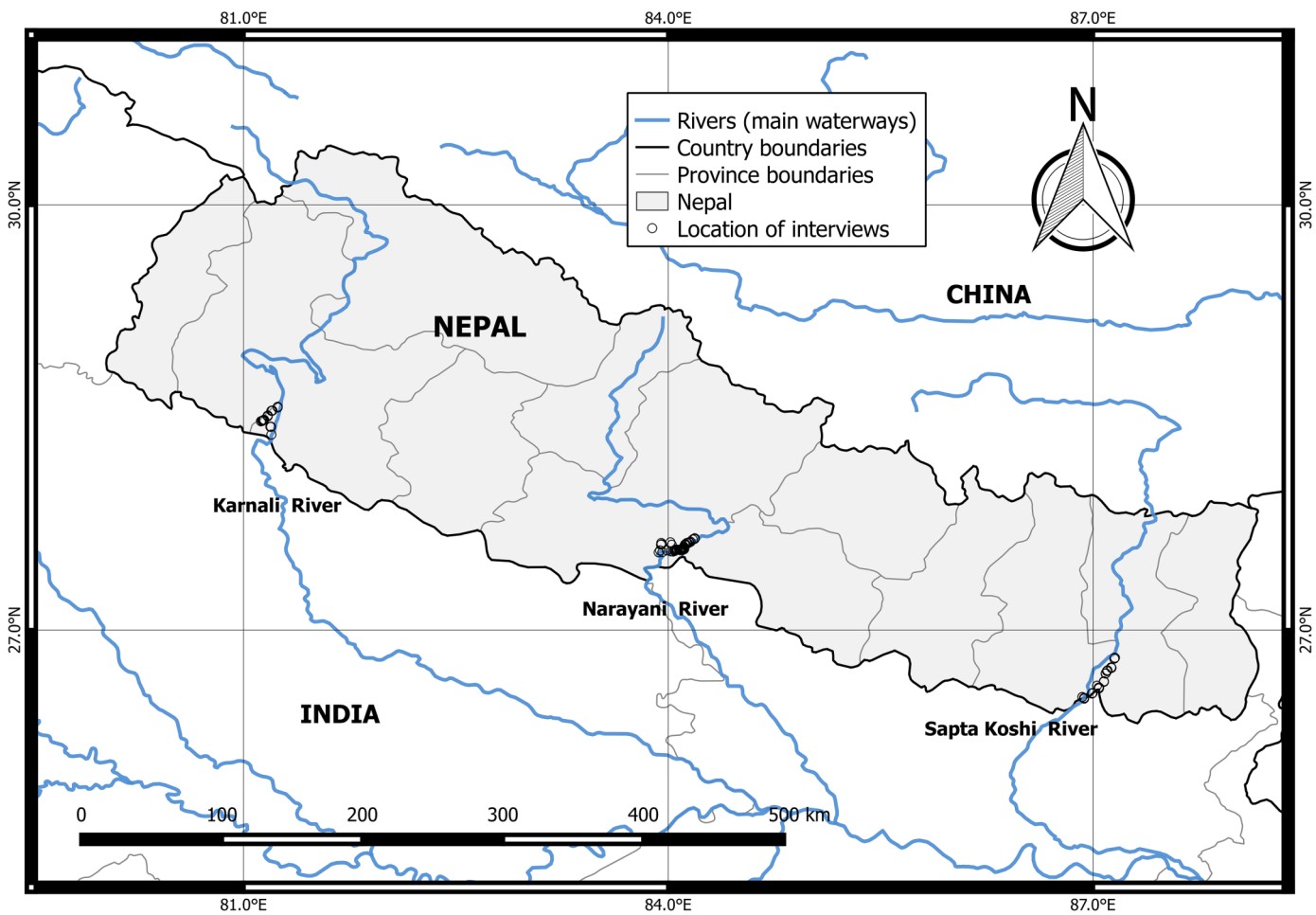

**Figure 1 Study area.** Map of interview locations in Nepal.

of artisanal fishermen that not only reside near the rivers, but regularly fish these rivers. The survey was administered by three technicians in the native Nepali language. To reduce any potential sampling bias, we randomly selected 15 percent of registered fishermen residing along the Karnali, Sapta Koshi and Narayani rivers to interview.

To increase the response rate and the quality of responses, the purpose and importance of the study was explained to fishermen before they were asked to participate in the survey. Also, the questionnaire format was clarified to each fisherman and then a point of contact for the study was provided to them. Overall, the questionnaire consisted of 87 simple and direct questions arranged into six themes: general description of fisheries, demographic information, fishing gear description, sightings and interactions with dolphins, dolphin population status, and preferred conservation measures. Questions were provided in open-ended and multiple-choice answer formats. To increase the response rate, demographic, general fishing information (i.e., fishing effort, gear, and experience), and fishermen attribute questions were asked at the beginning and more sensitive (income and interactions with dolphin) questions were asked at the end. Fishermen provided income information in Nepali currency, but we converted and reported their answers in US dollar

($1 USD = 98 NRs). Questions regarding dolphin interactions/sightings were divided by season (summer/winter) and time (past (>10 years) and present (<10 years)). The questions about potential threats and preferred conservation measures for the GRD in Nepal were provided in a multiple-choice style.

## Statistical analysis

Differences (expected vs observed) in categorical variables (e.g., demographics, fishery description, and fishermen perceptions of the dolphin population conservation status) between fishermen from the different rivers were tested using a Chi-square Goodness-of-Fit test ($\chi^2$). When expected cell frequencies were below 10, we used a Yates correction. We expected fishermen from each of the three river to answer every question similarly (null hypothesis; there was no significant difference between the frequency of expected and observed responses). To counter the effects of multiple paired testing (i.e., pair-wise comparisons), a $\chi^2$ approach was applied when differences among rivers were detected (*Todorov & Filzmoser, 2009*). The $\chi^2$ test was applied following the guidelines of *Koehler and Larntz (1980)*; k classes >3 (*Zar, 1994*). A Fligner-Killen test of homogeneity of variances (FK$\chi^2$) was applied for evaluating continuous variables (e.g., age, years living in the same village, fishing experience, fishing effort, and income). The FK$\chi^2$ test is an adaptation of the Kruskal-Wallis test that is robust against departures from normality (*Conover, Johnson & Johnson, 1981*). A Dunnett-Tukey-Kramer pairwise multiple comparison test was used to investigate the mean difference in more than two groups with unequal variance and sample size (*Lau, 2013*). A Mann-Whitney test was used to evaluate gillnet stretch mesh-size between the past (>10 years) and present (<10 years). Data were summarized, graphed, and evaluated using descriptive and hypothesis testing statistics. Data were managed using Microsoft Excel® and analyses were conducted using R version 3.0.2 (*R Core Team, 2013*). Statistical significance was defined as $P < 0.05$.

# RESULTS

## Survey

A total of 163 fishermen from the Karnali ($n = 56$), Sapta Koshi ($n = 47$) and Narayani ($n = 60$) rivers participated in the study. Each randomly selected fisherman was willing to participate and complete most of the questionnaire. Interviews with fishermen took between 15 and 107 minutes to complete, and the average time was $39.42 \pm 1.67$ minutes. A significant difference in interview time was detected among fishermen from the three rivers ($H = 124.03$; $P < 0.05$).

## Demographics

Fishermen ranged in age from 16 to 94 years of age, and the average age was 44.1 years of age. Fishermen from the Narayani river were significantly older than those from either the Karnali or Sapta Koshi rivers (Table 1). Eighty-seven percent of fishermen were men, but there were more women fishermen from the Narayani river than in the other two rivers. The fishermen represented 15 different ethnic groups, which were mostly Malha (27.0%),

**Table 1 Demographic characteristics of fishermen from the Karnali ($n$ = 56), Narayani ($n$ = 60), and Sapta Koshi ($n$ = 47) rivers.** Continuous data are shown as mean ± standard error and categorical data are shown as percentages. Differences between rivers and pairwise multiple comparisons were respectively tested with Fligner-Killeen and Dunnett-Tukey-Kramer test for continuous variables, and a Chi-square test with Yates correction (when required) was used for categorical variables. It should be noted that superscripts (a, b, c) sharing the same letter are statistically significantly different.

| Demographic characteristics | Total | Karnali river | Narayani river | Sapta Koshi river | Statistics, p-value |
|---|---|---|---|---|---|
| Age | 44.1 ± 1.1 | 38.7 ± 1.4[a] | 50.7 ± 1.8[a,b] | 42.1 ± 2.0[b] | FK$\chi^2$ = 6.3, p = 0.043 |
| Gender | | | | | |
|    Male | 86.5 | 87.5[a] | 75.0[a,b] | 100.0[b] | $\chi^2$ = 14.2, p = 0.001 |
|    Female | 13.5 | 12.5 | 25.0 | 0.0 | |
| Ethnicity | | | | | |
|    Bote | 16.6 | 0.0[a] | 45.0[a] | 0.0[a] | $\chi^2$ = 283.0, p < 0.001 |
|    Chaudhary | 11.0 | 10.7 | 18.3 | 0.0 | |
|    Malha | 27.0 | 0.0 | 0.0 | 93.6 | |
|    Sonaha | 25.2 | 73.2 | 0.0 | 0.0 | |
|    Other | 20.3 | 16.1 | 36.6 | 8.3 | |
| Education level | | | | | |
|    Illiterate | 69.4 | 82.1[a] | 80.0[b] | 42.6[a,b] | $\chi^2$ = 30.0, p < 0.001 |
|    Primary education | 22.7 | 8.9 | 15.0 | 48.9 | |
|    Secondary education | 6.8 | 7.1 | 5.0 | 8.5 | |
|    Higher education | 0.6 | 1.8 | 0.0 | 0.0 | |
| Permanent local resident | 93.9 | 96.4[a] | 86.7[a] | 100.0[a] | $\chi^2$ = 9.1, p = 0.011 |
| Years living in the same village | 43.6 ± 0.9 | 47.7 ± 1.1[a,b] | 41.8 ± 1.5[a] | 41.1 ± 2.0[b] | FK$\chi^2$ = 15.3, p < 0.001 |

Sonaha (25.2%), Bote (16.6%), and Chaudhary (11.0%). Most fishermen indicated they had little to no education; 70% reported to be illiterate and 22.7% had a primary education. The education level of fishermen was lowest in the Karnali river and highest in Sapta Koshi river. Most fishermen (93.9%) reported they had resided in their villages for over 40 years. Fishermen from the Karnali river stated they had resided longer in their villages than those from either the Narayani or Sapta Koshi rivers.

## Economics: Dependence on Fisheries

Reported earnings associated with fishing averaged $US 60.2 ± 2.6 per month; most fishermen (44.8%) earned less than $US 50 per month. Fishermen from the Karnali river indicated earning less money than fishermen from either the Narayani or Sapta Koshi rivers (Table 2). They also reported to us that they were highly dependent upon fishing for their income (78.5%), but they also stated having alternative sources of income, such as agriculture (47.9%). Monthly income from these alternative income sources ranged from $US 25 to $US 1,200, and the mean was $US 101.1 ± 9.9 per month. Overall, monthly earnings associated with alternative sources of income were lower in the Karnali river and higher in the Narayani river.

**Table 2 Characteristics of the fishing activity in the Karnali ($n = 56$), Narayani ($n = 60$), and Sapta Koshi ($n = 47$) rivers.** Continuous data are shown as mean ± standard error and categorical data are shown as percentages. Differences between rivers and pairwise multiple comparisons were tested with Fligner-Killeen and Dunnett-Tukey-Kramer test respectively for continuous variables, and a Chi-square test with Yates correction was used for categorical variables. It should be noted that superscripts (a, b, c) sharing the same letter are statistically significantly different.

| Fishing activity characteristics | Total | Karnali river | Narayani river | Sapta Koshi river | Statistics, p-value |
|---|---|---|---|---|---|
| **Fishing activity** | | | | | |
| Fishing is main occupation (%) | 78.5 | 75.0[a] | 70.0[b] | 93.6[b] | $\chi^2 = 9.3$, p = 0.009 |
| Years of experience fishing | 36.9 ± 1.1 | 35.5 ± 1.53[a] | 43.0 ± 2.0[a,b] | 30.7 1.5[b] | FK$\chi^2 = 17.7$, p < 0.001 |
| Age started fishing | 13.6 ± 0.3 | 15.2 ± 0.1[a] | 11.4 ± 0.5 [a,b] | 14.5 ± 0.7[b] | FK$\chi^2 = 35.8$, p < 0.001 |
| Occupation of father (%) | | [a] | [b] | [a] | $\chi^2 = 10.2$, p = 0.006 |
| Fisher | 77.9 | 75.0 | 31.7 | 93.6 | |
| Other | 22.1 | 25.0 | 68.3 | 6.4 | |
| **Fishing effort** | | | | | |
| Days fishing per week | 4.8 ± 0.2 | 5.0 ± 0.2[a] | 3.7 ± 0.3[b] | 6.2 ± 0.7[c] | FK$\chi^2 = 14.0$, p < 0.001 |
| Time spent fishing per day in winter (h) | 3.1 ± 0.1 | 2.8 ± 0.1[a] | 2.6 ± 0.2[b] | 4.1 ± 0.2[a,b] | FK$\chi^2 = 18.8$, p < 0.001 |
| Time spent fishing per day in summer (h) | 5.2 ± 0.2 | 3.7 ± 0.1[a] | 3.6 ± 0.1[b] | 9.0 ± 0.4[a,b] | FK$\chi^2 = 50.3$, p < 0.001 |
| Effective number of months fishing | 3.3 ± 0.1 | 2.6 ± 0.2[a] | 2.6 ± 0.1[b] | 5.1 ± 0.2[a,b] | FK$\chi^2 = 20.5$, p < 0.001 |
| **Economy** | | | | | |
| Monthly earnings from fishing ($) | 60.2 ± 2.6 | 26.0 ± 2.3[a,b] | 78.0 ± 3.7[a] | 78.2 ± 2.5[b] | FK$\chi^2 = 26.8$, p < 0.001 |
| Annual earnings from fishing ($) | 233.5 ± 16.3 | 84.0 ± 3.8[a] | 208.1 ± 18.0[a] | 418.6 ± 33.4[a] | FK$\chi^2 = 38.5$, p < 0.001 |
| Monthly earnings from other activities ($) | 101.1 ± 9.9 | 41.8 ± 2.0[a] | 171.0 ± 23.9[a] | 82.1 ± 3.5[a] | FK$\chi^2 = 32.2$, p < 0.001 |
| Secondary occupation | | [a] | [a] | [a] | FK$\chi^2 = 191.1$, p < 0.001 |
| Agricultural labor | 47.9 | 5.4 | 71.7 | 68.1 | |
| Gold filtering | 25.8 | 75.0 | 0.0 | 0.0 | |
| Fishing unbanned areas | 3.1 | 0.0 | 0.0 | 10.6 | |
| Daily wages | 9.8 | 0.0 | 26.7 | 0.0 | |
| Other | 10.4 | 17.9 | 1.7 | 10.7 | |

## Fishing activity

Seventy-eight percent of respondents reported fishing was their primary occupation. On average, fishermen had 36.9 ± 1.1 years of experience. Most fishermen indicated they began fishing at an early age; 88% percent reported they started fishing before the age of 15. Most fishermen (77.9%; Table 2) indicated their fathers were or currently are fishermen. Almost 65% of fishermen indicated they only owned one small wooden fishing vessel, but eight fishermen (4.9%) reported they owned more than one fishing vessel (Table 3). The mean fishing crew size was 4.7 ± 0.6 fishermen per vessel. The fishing crew size was significantly different among river segment ($H = 95.65$; $P < 0.05$).

## Fishing effort

The number of fishing days varied between 1 and 7 days per week, and the average (number of days per week fishermen spent fishing) was 4.8 ± 0.2 fishing days per

# PeerJ

**Table 3 Fishery description in the Karnali (*n* = 56), Narayani (*n* = 60), and Sapta Koshi (*n* = 47) rivers.** Continuous data are shown as mean ± standard error and categorical data are shown as percentages. Differences between rivers and pairwise multiple comparisons were tested with Fligner-Killeen and Dunnett-Tukey-Kramer test respectively for continuous variables, and a Chi-square test with Yates correction was used for categorical variables. It should be noted that superscripts (a, b, c) sharing the same letter are statistically significantly different.

| Fishing description | Total | Karnali river | Narayani river | Sapta Koshi river | Statistics, p-value |
|---|---|---|---|---|---|
| **Fishing boats** | | | | | |
| Owner of one boat | 64.8 | 82.1[a,b] | 52.5[a] | 59.6[b] | $\chi^2 = 11.8$, p = 0.003 |
| Type of boat | | [a] | [b] | [a,b] | $\chi^2 = 94.3$, p < 0.001 |
|   Single man traditional wooden boat | 81.0 | 100.0 | 100.0 | 17.9 | |
|   More than one man modern boat | 19.0 | 0.0 | 0.0 | 82.1 | |
| Average number fishermen per vessel | 4.7 ± 0.6 | 2.1 ± 0.1[a] | 11.8 ± 1.1[a,b] | 2.3 ± 0.1[b] | FK$\chi^2 = 26.8$, p < 0.001 |
| **Fishing gears** | | | | | |
| Fishing gear | | [a] | [a] | [a] | $\chi^2 = 23.8$, p < 0.001 |
|   Phekuwa Jaal | 25.8 | 14.3 | 3.3 | 68.1 | |
|   Maha Jaal | 24.5 | 71.4 | 0.0 | 0.0 | |
|   Pakhure Jaal | 22.7 | 0.0 | 58.3 | 2.3 | |
|   Other | 26.9 | 14.3 | 38.3 | 27.7 | |
| Net mesh size (cm) | 1.8 ± 0.2 | – | 1.7 ± 0.2[a] | 1.9 ± 0.2[b] | FK$\chi^2 = 0.1$, p = 0.990 |
| Net length (m) | 65.2 ± 6.7 | 170.2 ± 7.8[a,b] | 5.6 ± 1.2[a] | 14.1 ± 3.6[b] | FK$\chi^2 = 9.7$, p = 0.008 |
| Net width (m) | 4.6 ± 0.4 | 1.2 ± 0.1[a] | 9.1 ± 0.6[a] | 3.0 ± 0.1[a] | FK$\chi^2 = 55.1$, p < 0.001 |
| **Fishing time** | | | | | |
| Travel distance | 2.9 ± 0.1 | 2.6 ± 0.1[a] | 2.7 ± 0.2[a] | 3.3 ± 0.3[b] | FK$\chi^2 = 4.5$, p = 0.110 |
| Preferred fishing time (hrs) | 14:50 ± 0:16 | 15:52 ± 0:16[a] | 14:44 ± 0:32[b] | 13:44 ± 0:32[a,b] | FK$\chi^2 = 18.8$, p < 0.001 |
| Preferred fishing time | | [a] | [a] | [a] | $\chi^2 = 48.7$, p < 0.001 |
|   Breeding time for fish | 10.4 | 12.5 | 16.7 | 0.0 | |
|   High turbidity | 22.1 | 0.0 | 43.3 | 21.3 | |
|   Low water season | 65.0 | 85.7 | 36.7 | 76.6 | |
|   Summer season with hot water | 1.2 | 1.8 | 0.0 | 2.1 | |
|   Other | 1.2 | 0.0 | 3.4 | 0.0 | |

week. Seventy percent fished more than 4 days per week and 20.3% reported fishing one or two days per week. Overall, fishing effort varied significantly among river segment ($\chi^2 = 14.0$; $P < 0.001$). The highest fishing effort occurred in the Sapta Koshi river (6.2 ± 0.7 days/week) and lowest occurred in the Narayani river (3.7 ± 0.3 days/week). Overall fishing effort averaged 3.3 ± 0.1 months per year in all river systems, but it was significantly higher in the Sapta Koshi river than the other two rivers (Table 2). Fishing effort was significantly different between seasons ($P < 0.05$). In winter (dry season), fishermen spent 3.1 ± 0.1 hours/day fishing and in summer (wet season) they spent 5.2 ± 0.2 hours/day. This pattern was similar in the Karnali and Narayani rivers, but fishing effort in the Sapta Koshi river was significantly higher in summer and winter than in the Karnali ($H = 49.34$; $P < 0.05$) or Narayani rivers ($H = 94.78$; $P < 0.05$).

Most fishermen (90.2%) indicated they preferred to fish in the afternoon (1450 hrs ± 0.16), and during low water levels (65.0%; Table 3). The primary fishing period varied among river segment ($P < 0.001$). Fishermen from the Sapta Koshi river (13:44 ± 0.32) preferred to fish slightly earlier in the day than those from Narayani (14:44 ± 0.32) or Karnali rivers (15:52 ± 0.16). Fishermen also reported they preferred to fish during certain conditions. Most fishermen (>50%) from the Narayani and Sapta Koshi rivers stated they preferred to fish during high turbid and/or low water levels, while those from the Karnali river preferred to fish during the low water period.

## Fishing grounds

Fishermen indicated they usually fished close to their village. The mean distance travelled to the fishing grounds was 2.9 ± 0.1 km; fishermen rarely travelled more than 7 or 8 km. We did not detect a significant difference in the distance travelled upstream, but we did find that fishermen from the Narayani river travelled further downstream than those from either Sapta Koshi or Karnali rivers.

## Fishing gear

Fishermen reported using eight different types of fishing gear (Appendix 1). Twenty-five percent of fishermen used Phekuwa Jaal (cast net), 24.5% used Maha Jaal (gillnet), and 22.7% used Pakhure Jaal (cast net) fishing gear (Table 3). The other fishermen (27.8%) used a variety of nets, such as Bagaune Jaal (gillnet), Dadiya (cast net), Ghumauwa or Khaap Jaal (cast net), Paat or Hate Jaal (cast net), or Tiyari Jaal (gillnet). We found a significant difference in the type of fishing gear fishermen preferred to use among river segment ($\chi^2 = 23.80$ $P < 0.001$). Fishermen from the Narayani river primarily used Pakhure Jaal cast nets, whereas fishermen from the Karnali and Sapta Koshi rivers preferred to use Maha Jaal gillnets and Phekuwa Jaal cast nets, respectively.

Overall, the construction of gillnets used by fishermen varied in length, net depth, and stretch mesh-size. Gillnets ranged in length from 1.2 to 250 m. Sixty percent of fishermen reported using gillnets less than 10 m long, 30.1% were 10 and 100 m long, and another 30.1% used gillnets longer than 100 m; a few fishermen used more than one net. The average gillnet length was 65.2 ± 6.7 m. Fishermen from the Karnali river used longer gillnets than fishermen from either the Sapta Koshi or Narayani rivers ($\chi^2 = 9.7$; $P < 0.008$). Most fishermen (69.9%) stated the net depth was around 3 to 4 m; the mean net depth was 4.6 ± 0.4 m. A Chi-square test detected the net depth varied among river segment ($\chi^2 = 55.1$, $P < 0.001$). Fishermen from the Narayani river used gillnets that were deeper than those from either the Sapta Koshi and Karnali rivers (Table 3). The stretch mesh-size ranged from 0.23 to 7 cm, but the most common (79.8%) stretch mesh-size used by fishermen to construct their gillnets was around 2.0 cm or less. It should be noted that some fishermen (25.2%) indicated they recently changed to a smaller stretch mesh-size expecting to increase catch. A Mann-Whitney test showed there was a significant difference in the mean stretch mesh-size between the past and present ($P < 0.05$). Despite this change in the gear, they reported no major difference in catch.

## Fishing activity perceptions

Sixty-one percent of fishermen perceived a decline in catch over time and more than half believed the number of fishing boats in the area was similar to the past. Overall, perceptions about fishing activity (i.e., number of boats) were significantly different among fishermen from the three rivers ($\chi^2$ = 138.4; $P$ < 0.001). Most fishermen from the Karnali river believed there were fewer fishing boats now that before, while fishermen from the other two rivers did not think there was a difference. Fishermen from the Karnali and Sapta Koshi rivers also believed fishing was worse now than before. In contrast, most fishermen from the Narayani river (70.0%) actually thought fishing was better now than before. Interestingly, every fisherman stated they did not believe fishing was a good job and preferred their children pursued another occupation. Some fishermen (35.0%) indicated they wanted their children to work for a private firm followed by a government agency (31.3%) or a non-government organization (12.3%) (Table 4).

## Ganges river dolphin sightings and observations

Most fishermen (62.6%) indicated they rarely spotted GRD on recent fishing trips, but many (60.7%) reported regularly spotting them in the past (>10 years). Fishermen from the Karnali river indicated they occasionally spotted GRD on recent fishing trips, while most fishermen from the Narayani and Sapta Koshi rivers reported they seldom spotted them ($\chi^2$ = 70.4; $P$ < 0.001). Karnali river fishermen reported occasionally spotting GRD in the past, while Narayani and Sapta Koshi river fishermen reported frequently spotting them in the past. Karnali river fishermen reported they used to spot around two GRD in the past, while Sapta Koshi and the Narayani river fishermen indicated spotting four or more individuals, respectively.

In general, most GRD were spotted in deep pool areas and most were observed diving. A Chi-square test detected a significant difference in the location where fishermen spotted GRD among river segments ($\chi^2$ (4, 167) = 106.39; $P$ < 0.05). While every fisherman from the Narayani river, and most from the Karnali river reported spotting GRD in deep pools, Sapta river fishermen indicated they usually spotted them in the confluence and main channel areas. Fishermen from the Narayani and Sapta Koshi rivers reported spotting GRD actively diving, while those from Karnali river indicated they often spotted only their back and/or snout at the surface. Of the 163 fishermen interviewed, only one from the Narayani river reported he had encountered a dead GRD.

## Ganges river dolphin conservation measures

Most fishermen (89.5%) perceived the GRD population had declined over time for a variety of reasons. A Chi-square test detected a significant difference in the observed and expected counts in the reasons why fishermen perceived the GRD population had declined ($\chi^2$ (12, 177) = 140.12; $P$ < 0.05). Most fishermen believed the main threat to GRD were humans, stating the construction of dams/irrigations systems (53.5%) and fishing were the main reasons the GRD population had declined. Some fishermen (32.1%) thought the recent decline in the GRD population was associated with physical changes (width and

**Table 4 Fishermen perception about the fishing activity and fisheries as a job in the Karnali ($n = 56$), Narayani ($n = 60$), and Sapta Koshi ($n = 47$) rivers.** Continuous data are shown as mean ± standard error and categorical data are shown as percentages. Differences between rivers and pairwise multiple comparisons were tested with Fligner-Killeen and Dunnett-Tukey-Kramer test respectively for continuous variables, and a Chi-square test with Yates correction was used for categorical variables. It should be noted that superscripts (a, b, c) sharing the same letter are statistically significantly different.

| Fishermen perceptions and opinions | Total | Karnali river | Narayani river | Sapta Koshi river | Statistic, p-value |
|---|---|---|---|---|---|
| **Perception about fishing** | | | | | |
| Perception about changes in the amount of fish caught over time | | [a] | [a] | [a] | $\chi^2 = 138.4$, p < 0.001 |
|   Worse than before | 61.3 | 100.0 | 6.4 | 66.1 | |
|   Same as before | 18.4 | 0.0 | 23.4 | 33.9 | |
|   Better than before | 20.2 | 0.0 | 70.2 | 0.0 | |
| Perception about changes in the quantity of boats in the river | | [a] | [a] | [a] | $\chi^2 = 89.4$, p < 0.001 |
|   Fewer than before | 36.8 | 78.3 | 14.9 | 10.7 | |
|   Same as before | 54.0 | 10.0 | 68.1 | 89.3 | |
|   More than before | 9.2 | 11.7 | 17.0 | 0.0 | |
| **Fishing job** | | | | | |
| Don't want their children to be a fisher | 100.0 | 100.0[a] | 100.0[b] | 100.0[c] | $\chi^2 = 1.6$, p = 0.442 |
| Don't think fishing is a good job | 100.0 | 100.0[a] | 100.0[b] | 100.0[c] | $\chi^2 = 1.6$, p = 0.442 |
| Which job they would like for their children | | [a] | [a] | [a] | $\chi^2 = 99.3$, p < 0.001 |
|   Agriculture | 10.4 | 1.8 | 21.7 | 6.4 | |
|   Fishing business | 3.7 | 3.6 | 0.0 | 8.5 | |
|   Governmental job | 31.3 | 10.7 | 51.7 | 29.8 | |
|   NGO | 12.3 | 3.6 | 11.7 | 23.4 | |
|   Private firm | 35.0 | 80.4 | 5.0 | 19.1 | |
|   Other small business | 7.4 | 0.0 | 10.0 | 12.8 | |

depth) in the river (Table 5); most fishermen from the Karnali and Narayani rivers believed the decline in the GRD was associated with low water conditions.

Favorably, our study revealed that the conservation of the GRD seemed to be important to every fisherman. Actually, most fishermen suggested that increasing GRD awareness and establishing new training opportunities using locally available natural and social resources would help reduce fishing pressure and risk to GRD. Seventy percent of fishermen thought it was possible to develop eco-tourism in Nepal. Karnali and Sapta Koshi river fishermen indicated they wanted eco-tourism; however, many Narayani river fishermen were opposed to the idea. Of the fishermen that wanted to be re-trained, almost half of them chose masonry or carpentry professions.

## DISCUSSION

Anthropogenic activities (e.g., commercial fishing and vessel collisions) are the leading cause of mortality for most cetaceans around the world (*van der Hoop et al., 2013*). Some cetacean injuries and mortalities are associated with vessel strikes and other human-induced activities (*Silber et al., 2015*); however, many injuries and mortalities are attributed to the incidental entanglement with fishing gear, especially monofilament

**Table 5 Fishermen perceptions about dolphin population and conservation status in the Karnali ($n = 56$), Narayani ($n = 60$), and Sapta Koshi ($n = 47$) rivers.** Continuous data are shown as mean ± standard error and categorical data are shown as percentages. Differences between rivers and pairwise multiple comparisons were tested with Fligner-Killeen and Dunnett-Tukey-Kramer test respectively for continuous variables, and a Chi-square test with Yates correction was used for categorical variables. It should be noted that superscripts (a, b, c) sharing the same letter are statistically significantly different.

| Perceptions about dolphins and their conservation | Total | Karnali river | Narayani river | Sapta Koshi river | Statistic, p-value |
|---|---|---|---|---|---|
| **Dolphin sightings** | | | | | |
| Does not know (saw or heard) of dead dolphins | 99.4 | 100.0[a] | 98.3[b] | 100.0[c] | $\chi^2 = 1.7$, p = 0.422 |
| Perceives to seeing dolphins often in the past | 61.3 | 28.6[a,b] | 73.3[a] | 85.1[b] | $\chi^2 = 53.5$, p < 0.001 |
| Perceives to rarely see dolphins now | 62.6 | 23.2[a] | 98.3[a] | 63.8[a] | $\chi^2 = 70.4$, p < 0.001 |
| Type of habitat where dolphins are most often sighted | | [a] | [a] | [a] | $\chi^2 = 104.7$, p < 0.001 |
|   Deep pool (depth >3m) | 56.0 | 50.0 | 100.0 | 10.6 | |
|   Confluence | 12.6 | 7.1 | 0.0 | 34.0 | |
|   Straight channel (depth <3m) | 26.4 | 42.9 | 0.0 | 38.3 | |
|   Meandering | 5.0 | 0.0 | 0.0 | 17.0 | |
| Type of behavior when dolphins are sighted | | [a,b] | [a] | [b] | $\chi^2 = 138.2$, p < 0.001 |
|   Diving | 66.5 | 7.1 | 100.0 | 100.0 | |
|   Showing back and snout | 31.6 | 87.5 | 0.0 | 0.0 | |
|   Swimming | 1.9 | 5.4 | 0.0 | 0.0 | |
| Distance from dolphin to boat during sightings (m) | 48.1 ± 8.4 | 1.8 ± 0.1[a] | 131.4 ± 19.3[a,b] | 3 ± 0.0[b] | FK$\chi^2 = 74.8$, p < 0.001 |
| **Dolphin conservation** | | | | | |
| Perceives decrease in number of dolphins over time | 89.5 | 87.5[a] | 100.0[a,b] | 78.7[b] | $\chi^2 = 13.0$, p = 0.002 |
| Perceived major threats to dolphins | | [a] | [a] | [a] | $\chi^2 = 64.7$, p < 0.001 |
|   Habitat overlapped with fishermen | 10.7 | 0.0 | 28.3 | 0.0 | |
|   Low depth and width of river | 32.1 | 12.5 | 36.7 | 51.2 | |
|   High human disturbances | 53.5 | 85.7 | 26.7 | 48.8 | |
|   Decrease in prey density | 3.7 | 1.8 | 8.3 | 0.0 | |
| Ways to conserve dolphins | | [a,b] | [a] | [b] | $\chi^2 = 64.3$, p = 0.001 |
|   Awareness among the fishermen/river dependent communities | 53.4 | 89.3 | 30.0 | 40.4 | |
|   Enterprise training facilities for river dependents | 23.3 | 1.8 | 38.3 | 29.8 | |
|   Monitoring of fishing activities through watch group | 8.6 | 3.6 | 13.3 | 8.5 | |
|   Punishing people engaged in illegal activities according to law | 5.5 | 0.0 | 5.0 | 12.8 | |
|   Careful fishing by avoiding killing dolphins | 4.9 | 5.4 | 1.7 | 8.5 | |
|   Other | 4.3 | 0.0 | 11.7 | 0.0 | |

gillnets (*Reeves, McClellan & Werner, 2013*). Regrettably, limited information is available describing cetacean bycatch in gillnets, especially fishery interactions with freshwater cetaceans. *Reeves, McClellan & Werner (2013)* stated that understanding fishery interactions is essential for preventing further losses of cetacean diversity and abundance, particularly in remote regions. In Nepal and India, the incidental entanglement of GRD with fishing gear is one of the major threats to the conservation and recovery of the GRD (*Wakid & Braulik, 2009*; *Kelkar et al., 2010*; *Sinha, Behera &*

*Choudhary, 2010*). Developing and implementing effective recovery actions for the GRD requires managers having adequate socio-economic and fishery information. Without this type of information, it is almost impossible for conservation managers to make informed and effective decisions. Given the economic constraints of researchers in Nepal, in terms of available research funding, information describing artisanal fisheries and potential conservation implications for the GRD has been unavailable until now.

## Demographics and economics

Interviews revealed that established communities and associated ethnic groups (e.g., Malaha, Sonaha, Bote, and Chaudary) residing (<1 km) along major rivers in Nepal rely almost exclusively on fishing for their income. Fishing has not only been a way of life for many residents since an early age (~ 15 years old), but most fishermen fish for most of their lives. In fact, we discovered that most fishermen began fishing at an early age and continued to fish throughout their life, which limited their educational opportunities and ability to pursue other occupations. Despite the importance of fishing to the community, we were surprised to know that most fishermen did not want their children to pursue fishing as a job. Given this strong belief, we believe it is possible, with the right training, that parents could encourage their children to pursue other occupations, particularly since some of them already have a second job, such as agriculture. Obviously, reducing the fishing pressure in the region would have a positive impact on the GRD even though the construction of dams and other anthropogenic activities are still a major problem for GRD. Alternative income opportunities for river-dependent residents in Nepal are clearly limited, but there are still a few options that could benefit locals and the GRD, such as eco-tourism, farming, or simply changing fishing tactics or fishing gear. We are aware the farming trade is growing throughout Nepal (*Joshi, Conroy & Witcombe, 2012*), so it is possible that Nepalese fishermen would consider permanently changing occupations.

According to the *FAO (2011)*, Nepal was the 12th poorest country in the world during 2010 with a per capita income of US $480. Although employment opportunities are limited, the economic status in Nepal is improving, which could give fishermen other options for making a living in the near future. Agriculture (paddy, maize, wheat, millet, and legumes) is a large industry in Nepal, but there are other non-agricultural industries that provide jobs, such as manufacturing, construction, and personal services (CBS, 2011). Unfortunately, these options are limited in rural areas (river communities) so fishermen have less economic opportunities. Based on our interviews, we know fishermen would be interested in establishing some sort of ecotourism, which is possible for Nepal. Actually, tourism is already a major industry (US $170 million annually) in various regions of Nepal, so expanding this industry could help reduce poverty in both urban and rural areas (*GON, 2013*). According to *Chan & Bhatta (2013)*, tourism contributes to about 7.4 percent of Nepal's National gross domestic product and 5.8 percent of the total employment. A study by *GON (2013)* reported most tourists are from India, China, Sri Lanka, United States, and the United Kingdom. The report highlighted that most tourists travel to Nepal for holiday/pleasure, and enjoy visiting National Parks and Wildlife

Reserves. Thus, it is highly likely that Nepal could develop an ecotourism industry in rural areas, but to do it correctly it will take a lot of planning and support from various groups (government institutions, NGOs, and private companies), especially since infrastructure will need to be developed in these remote locations (*Chan & Bhatta, 2013*). Ecotourism has already been successful in various remote locations, such as India, Belize, and the Dai villages of Yunnan Province of China (*Chan & Bhatta, 2013*). Maybe expanding ecotourism would provide other job options for fishermen while at the same time provide a way to promote the conservation and recovery of the GRD in Nepal.

## Fishing activity

Most fishermen only own one fishing vessel, so it appears that local river residents are simply attempting to support their families rather than establishing large thriving fishing businesses with a fleet of vessels. Our findings suggest that fishing is probably not expanding in some regions of Nepal, but additional research is warranted. According to responses, the mean crew size is between 4 and 5, but fishermen from the Narayani river tend to use larger crews because many of them cannot purchase their own vessel. Assuming a larger crew corresponds to less gear in the water then overall risk to GRD could be relatively lower in the Narayani river than the other two rivers.

Our survey revealed that fishermen from the Narayani river preferred to use cast nets rather than gillnets, which reduces the risk for GRD. Bycatch associated with gillnets is a major issue for cetaceans worldwide (*Kennelly & Broadhurst, 2002*). Thus, switching from gillnets to cast nets might be a viable option for fishermen from Karnali and Sapta Koshi river, especially since Sapta Koshi fishermen reported they thought fishing was better now than before. We should point out that we did not segregate data by age-class, so it is possible that younger fishermen have a different or skewed perception about fishing than older fishermen. The successful transition into using cast nets rather than gillnets will depend on the target catch since some species of fish do not display schooling behavior; schooling fish are much easier to target with cast nets than gillnets. We should also point out that the fishermen's perception that fishing is better now than before could potentially intensify localized fishing pressure and increase the risk to GRD inhabiting the Sapta Koshi river. The GRD population in the Sapta Koshi river has been declining at an alarming rate over the last 25 years, so additional fishing pressure poses an immediate risk to the conservation of the species, particularly since immense fishing pressure is still a problem in the Sapta Koshi river (*Chaudhary, 2007*).

In the Narayani river, fishermen reported they believed fishing was worse now than before. Assuming this is an accurate description of the situation; fishermen could be directly competing with the GRD by taking fish that are essential to the GRD diet. Given the limited fishery resources, fishermen could be indirectly impacting the GRD in the Narayani river. This scenario has been reported by researchers in other regions. For instance, *Secchi & Wang (2002)* reported that gillnet fishermen in Brazil have indirectly impacted the diet of Franciscana (*Pontoporia blainvillei*), which is one of the endangered cetaceans. Is this situation occurring in Nepal? We recommend future studies to investigate this potential phenomenon.

## Fishing effort

Fishermen depend on fishery resources to support their families, so most of them fish as much as possible (>4 days per week). Interestingly, fishermen from the Sapta Koshi river reported they fish every day, which likely increases the risk to the GRD in that region. These same fishermen also reported they preferred to fish in the morning rather than in the afternoon, which is the opposite tactic used by fishermen from either the Narayani or Karnali rivers. We are unaware why there were differences in preferred fishing periods, but it could be related to target species. Despite the reasons, this fishing tactic poses a risk to GRD. Based on recent research, (e.g. *Sinha, Behera & Choudhary, 2010*; *Sasaki-Yamamoto et al., 2013*) it appears GRD are more active in early-morning (08:00–11:00 hrs) and late-afternoon (13:30–16:00 hrs). Unfortunately, our study revealed that fishermen also preferred to fish during these periods, which poses a great risk to the GRD. Given this situation, it is probable that GRD are depredating and interacting with gillnets; depredation in gillnets is a common behavior for many cetaceans around the world (*Read et al., 2003*; *Mathias, 2012*; *Waples et al., 2013*). Additional research is warranted, but it might be possible that Nepalese fishermen could set their gear during the day (1100–1330 hrs) instead of the morning and late-afternoon without compromising their catch?

The GRD migrates seasonally according to water level (dry vs wet season). *Kelkar et al. (2010)* & *Paudel et al., (2015)* all reported that GRD are found in deep pools or the main channels of rivers in the dry season (October–May), and migrate upstream to tributaries following the monsoon period (June–September). Seasonal movement in conjunction with the low water period has also been reported for GRD in the Brahmaputra river from the Assam-Arunachal to India-Bangladesh border (*Wakid & Braulik, 2009*). Given these movement patterns, fishing in winter during low water season seems to pose a greater risk to the GRD since they are more concentrated in specific areas like deep pools where fisherman prefer to fish; assuming more gear is set in pools than along banks. Although interviews revealed that fishermen spent almost twice as many hours fishing in summer (5.7 hours/day) than in winter (3.7 hours/day), fishing in winter still seems to pose a risk to GRD. Regardless of the season, most fishermen reported they preferred to fish in tributaries, especially in the Karnali river. It should be noted that fishing in Karnali river area appears to threaten GRD during the wet season because the Karnali and Sapta Koshi rivers are more critical to GRD population than the Narayani river population given their lower relative abundance (*Paudel et al., 2015*). Even though relative abundance is generally lower (*Kelkar et al., 2010*; *Paudel et al., 2015*) in the post-monsoon than the pre-monsoon period (*Paudel et al., 2015*), fishing in the dry season could also endanger the GRD because the lower water level makes it more difficult for the GRD to avoid being entangled in gillnets. Our study revealed that the average net depth used by fishermen was 4.5 m, which also corresponds to the average water depth (4.4 m) where GRD are usually spotted (*Paudel et al., 2015*). Because the net depth is greater than the average water depth of many river sections during the dry season, this situation suggests this is a major danger for the GRD.

In our opinion, the proximity to the fishing grounds also likely poses a serious threat to the GRD. Based on interviews, fishermen indicated that almost all of them set their nets within 5.4 km of their village (2.9 km upstream or 2.5 km downstream). Given this tactic, it appears that nets are concentrated in specific areas (i.e., fishing hotspots), which could reduce the mobility for the GRD and increase the risk of being accidentally entangled. More nets in specific areas have been shown to increase the risk to marine mammals (e.g., *Kinsas, 2002*). In addition, it is likely that GRD are attracted to these fishing hotspots because they commonly depredate catch from nets. According to *Chaudhary (2007)*, a hotspot for the GRD is the southern section of the Koshi barrage, which is also a popular fishing spot. Spatial overlap between GRD distribution and fishing activity was previously reported by *Malla (2009)* and *Kelkar et al. (2010)*. *Smith (1993)* indicated that the primary habitats of GRD also coincide with the areas of greatest human use. Interestingly, interviews with Narayani River fishermen revealed they tend to travel further downstream, which suggests that they are expanding their fishing range. Expanding the fishing range could either be increasing or reducing the risk to GRD in the Narayani River. Additional research is warranted.

## Fishing gear

Fishermen use a variety of monofilament gillnets and cast nets, but we did find some differences in fishing gear among river segment. Fishermen from the Narayani and Sapta Koshi rivers preferred to use cast nets, whereas fishermen from the Karnali River primarily used gillnets. Plainly, cast nets pose a lower risk to the GRD than gillnets given their smaller size and the deployment method. Cast nets are thrown off a vessel and immediately retrieved, whereas gillnets are allowed to soak for an extended period; soak time and cetacean entanglement are positively correlated (*Rossman & Palka, 2011*). It is difficult to understand why most fishermen from the Karnali river are inclined to use gillnets instead of casts, but it is probably associated with the target species. We recommend additional research to understand fishing tactics and gear in the Karnali River.

Although most fishermen reported using gillnets less than 10 m long, 30% stated they used gillnets longer than 100 m; net length and cetacean-fishery interactions are generally positively correlated. Besides net length, soak duration is also a potential problem for GRD. We don't know much about the soak time, but this could be a major risk issue for GRD, especially if fishermen soak their nets overnight. The length of gillnet and cetacean entanglement risk is probably correlated, but is difficult to predict what factor increases the probability of entanglement. Interviews pointed out that gillnet length varied significantly by river segment. Overall, fishermen from the Karnali river used longer gillnets than fishermen from either the Sapta Koshi or Narayani rivers. Again, we do not know why this was the case, but understanding this tactic could help us recommend alternatives to fishermen that might reduce the risk to GRD in the Karnali river. Despite the fact that fishermen from the Narayani river used shorter gillnets, they reported their gillnets were much deeper than those used by fishermen from either the Karnali or Sapta Koshi rivers. Regrettably, using deeper nets could actually be more harmful to the GRD
than longer nets since the GRD is known to chase prey along the bottom (*Sinha, Behera & Choudhary, 2010*).

The majority of fishermen used gillnets constructed with a stretch mesh-size less than 2.0 cm. We also observed that fishermen continued to construct nets with smaller stretch mesh over the years, which suggests that catch is decreasing over time. Because gillnets are selective, stretch mesh-size is an important factor to evaluate since it relates to catch composition and size-frequency. The type and size of catch could be negatively impacting the GRD diet; GRD prey on Reba carp (*Cirrhinus reba*) and Baam (*Mastacembelus armatus*)(*Bashir et al., 2010*). In the Vikramshila Gangetic Dolphin Sanctuary (a 65-km stretch of the Ganga River between Sultanganj and Kahalgaon towns in Bhagalpur, Bihar, India), *Kelkar et al. (2010)* discovered that the size distribution of fish were mostly (75%) within the size range preferred by GRD. These finding suggests that fishermen are affecting the GRD diet in India.

Maybe local officials should consider implementing gillnet mitigation measures to reduce the entanglement risk for GRD, such as acoustic deterrents (*Dawson et al. 2013*)? Various mitigation options have been used before in the other regions to reduce the frequency of marine mammal-fishery interactions, such as changing human behavior (time-area closures) and gear modifications (mesh-size, gillnet length, soak time, and tie-downs). We recommend funding research to investigate gear modifications, and suggest that fishermen start using best management practices, such as reduced soak times or continuous monitoring of nets. We suggest removing entangled fish on a regular basis could potentially reduce GRD depredation and overall risk.

## Ganges river dolphin sightings and observations

Based on responses, fishermen spot fewer GRD now than before; thus, it appears the GRD continues to decline in Nepal river systems – a finding that is consistent with previous studies (*Smith, 1993*; *Reeves, Smith & Kasuya, 2000*; *Reeves et al., 2003*; *Paudel et al., 2015*). Little is known about the social aspects of the GRD, but it is likely that smaller group sizes, including reports of single individuals are indicative of the fragmentation of the population as a whole and habitat degradation. Small groups lack the benefits associated with social living (e.g., predator avoidance, detection of prey, and facilitated reproductive activities) (*Baird & Whitehead, 2000*). Fishermen also indicated that fewer GRD were seen in the Narayani and Karnali rivers than in the Sapta Koshi, which is consistent with previous research (*Paudel et al., 2015*). *Paudel et al. (2015)* reported that the GRD range is shrinking and fewer dolphins are using the remaining available habitat in the Karnali river system, which suggests the population may not be able to recover (*Smith, 1993*; *Paudel et al., 2015*).

## Ganges river dolphin conservation measures

Most fishermen believed the threat of the GRD is related to water pollution, and/or dam/ irrigation development. The construction of dams and other water diversion projects for hydro-electric power production and irrigation lowers local water levels not only permanently alters river ecology, but it leads to a smaller GRD range and changes the daily

and seasonal movement patterns. Obviously, water level is an important habitat factor that controls the seasonal distribution of GRD since this species have never been observed in water levels less than 2.0 m (*Biswas & Boruah, 2000*; *Braulik et al., 2012*; *Paudel et al., 2015*). Construction of dams in Nepal is likely to continue since only about 50% of urban and 5% of the rural population has access to electricity (*Bergner, 2012*). The construction of dams in Nepal also negatively impacts GRD habitat and causes population fragmentation. Water flow diversion by the construction of a barrage during the dry season has even led to the stranding of a GRD (*Smith & Braulik, 2012*). The construction of dams in Nepal is serious situation. In fact, *Smith & Reeves (2000)* stated that building a high dam in the Karnali river would "almost certainly eliminate the small amount of dolphin habitat in Nepal's last river with a potentially viable dolphin population". The same scenario is found in the Sapta Koshi river, where the Koshi barrage, above 7 km from Nepal/India boarder, deters the upstream movement of river dolphin during summer season.

## CONCLUSIONS

The GRD is recognized as one of the most endangered cetaceans in the world. In Nepal, its distribution is restricted to the Narayani, Sapta Koshi, and Karnali river systems. Regrettably, various anthropogenic activities continue to jeopardize the GRD's survival, such as artisanal fishing. Nepal is one of the poorest countries in the world, so economic opportunities are limited, especially in rural remote areas. Although river-dependent residents residing along the Narayani, Sapta Koshi, and Karnali rivers have other sources of income, artisanal fishing is their main occupation. Based on interviews with local fishermen, it appears there is spatial overlap between the fishing grounds and potential GRD suitable habitat. This spatial overlap between fisheries and GRD potentially increases the risk of GRD-fishery interactions and threatens the recovery of the species in Nepal, especially since most fishermen reported using monofilament gillnets. Although we did not directly sample catch, artisanal fisheries could be indirectly impacting the GRD's diet by taking preferred prey. We recommend additional research into this topic. The GRD and fishery interaction problem in Nepal is challenging to solve given the socio-economic situation, but fishing gear modifications (mesh-size, gillnet length, soak time, and tie downs), changing human behaviour (time-area closures), and switching professions (eco enterprise business using natural and socio economic resources) are a few options that have been explored in other regions to reduce fishery interactions with marine mammals. For instance, *Hall (2009)* stated that gillnet gear characteristics affect target catch and bycatch so it is important to understand the following: (1) the way gillnets capture species (e.g., gilling and entangling); (2) whether gillnets are fixed or drifting; (3) where in the water column gillnets are set (surface, mid-water, or bottom); (4) mesh size; (5) type of construction materials; and (6) hung ratio. However, before any type of mitigation measures can be implemented, we must first understand the fishery characteristics, especially information on the gear and target catch. As such, we recommend conservation managers fund a study to thoroughly evaluate the artisanal fishery in Nepal. Lastly, we believe conservation managers need to seriously consider using

the non-transboundary management approach with neighbouring countries to protect the remaining GRD population before it's too late.

## ACKNOWLEDGEMENTS

We are grateful to S. Basnet and D. Nath for collecting field data. We also thank P. Basnet, M. Haiju and S. Raut for data entry, and G.J. Pierce from the University of Aberdeen for providing help with the analysis of socio-economic data. We especially thank G. Silber and J. Gearhart from the National Marine Fisheries Service for providing valuable edits and recommendations that greatly improved the article. We also thank the anonymous reviewers for providing critical comments that improved the article. Lastly, we thank the Kathmandu Forestry College for providing sufficient time to conduct this project.

### Funding

Funding was received from the Ocean Park Conservation Foundation, Hong Kong and Rufford Foundation, UK, and from IMATA, USA. The funders had no role in study design, data collection and analysis, decision to publish, or preparation of the manuscript.

### Competing Interests

Juan C. Levesque is an employee of Environmental Resources Management. The authors declare that they have no competing interests.

### Author Contributions

- Shambhu Paudel conceived and designed the experiments, performed the experiments, analyzed the data, contributed reagents/materials/analysis tools, wrote the paper, prepared figures and/or tables, reviewed drafts of the paper.
- Juan C Levesque analyzed the data, contributed reagents/materials/analysis tools, wrote the paper, prepared figures and/or tables, reviewed drafts of the paper.
- Camilo Saavedra analyzed the data, contributed reagents/materials/analysis tools, wrote the paper, prepared figures and/or tables, reviewed drafts of the paper.
- Cristina Pita contributed reagents/materials/analysis tools, wrote the paper, reviewed drafts of the paper.
- Prabhat Pal contributed reagents/materials/analysis tools, wrote the paper, reviewed drafts of the paper.

### Human Ethics

The following information was supplied relating to ethical approvals (i.e., approving body and any reference numbers):

Department of National Parks and Wildlife Conservation (DNPWC) provided the approval to undertake this study in three river systems of Nepal: Reference number 353.

## Data Deposition

The raw data is provided as Supplemental Dataset files.

## Supplemental Information

Supplemental information for this article can be found online at http://dx.doi.org/10.7717/peerj.1563#supplemental-information.

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
