# Peer review of "Characterization of the artisanal fishing communities in Nepal and potential implications for the conservation and management of Ganges River Dolphin (Platanista gangetica gangetica)"

_PeerJ, doi:10.7717/peerj.1563_

## Round 0.1 · original submission · Major Revisions

Please consider all the suggestions made by the 3 reviewers in your revised version.

Reviewer 1 ·

Basic reporting

The article describes the fishery and socio-economic context of fisheries in Nepal, where the endangered Ganges River dolphin is found. The data is based on interviews with 29 local fishermen in river systems in Nepal where the dolphin is found.
Figure 1 would benefit from adding the distribution go the GRD (if known)

Experimental design

The methods require further details on how the interviews were conducted.
It would be useful to include/compare to national statistics for Nepal (for example published by OECD), to understand how the fishers compare to the national economy. This issue is not brought up until late in the discussion.

Validity of the findings

Authors should check their results (ie. perceptions of the state of the fishery and dolphin population) for differences between age groups. The results could be confounded by 'shifting baselines' where younger generations have positive perceptions because they have no awareness of what has been lost (i.e. line 375).
The discussion has several places with speculative statements, that need to be clearly marked as such. The discussion requires a more in depth analysis of the results and a better review of existing literature.
The conclusions are not supported by the findings.
No mention is made of data deposited in an archive or made available.

Additional comments

Introduction.
Line 55, "obligate cetacean", would imply that there are facultative cetaceans that choose to be cetaceans.
Lines 65-66. This section makes it seem as if gillnet bycatch is not an issue, then lines 81-88 actually explain the conflict with fisheries (and then the Discussion in Line 325 makes it seems as if its the largest source of mortality). These two paragraphs could be merged and better structured.
Methods
Please mention the language of the survey, who interviewed the participants, and if it was the same interviewees in each community.
Line 172-173. Instead of saying "many years", just keep the mean in the second part of the sentence.
Economics: Where the fishers asked about dollar amounts or local currency? what was the exchange rate used to estimate the US amounts.
Line 178-179. Sentence is misleading. If 45% of fishermen earned less than 50 US per month, then 55% earned more than 50. Thus a majority of fishermen earned more than $50US per month.
Line 221, should be "slightly fewer", not less.
Line 220-226. Could you include a table or a description of the differences between the fishing nets? for example both the Pakhure Jall and Dadiya are cast nets, but how are they different? Different gears could have different bycatch rates.
Lines 239-240. "differences between what they sue now and before". Before what?
Line 258. Are there different perceptions by age group? do you detect a 'shifting baseline' syndrome?
Discussion
Lines 350-363. A comparison between Nepal and the US is not meaningful, both in the context and difference in economies between the two countries. Use sites with more comparable situations. For example, the situation of the vaquita (Phocoena sinus) in Mexico, where a variety of economic incentives have been used to try to move fishers to alternative occupations.
Lines 373-374. Without understanding the differences between gears, and why fishers prefer one over the other, this statement is speculative.
Lines 397-399. Speculative
Line 410. likely poses
Line 437. Is there more information available on the differences between gears described in the paper?
Lines 443-445. This is speculative, the authors could find more information on the differences between casting and gillnet regarding efficiency, effort, and amount caught from the literature.
Line 465. Is there any data on catch from these rivers?
Titles of the tables are confusing because they all start with "Demographic characteristics", maybe better to use, demographic characteristics, fishing characteristics, characteristics of the fishery

·

Basic reporting

The paper is well written and organized. There is one typographical error at line 82. The citation Sinha et al., 2010 needs a closed parenthesis. In addition, the authors refer to net depth (width) in the text and list net width in Table 3. I suggest using the traditionally used gill net gear descriptor "net depth" both in Table 3 and in the text.

Experimental design

The study design appears to be sound. No comments.

Validity of the findings

No Comments

Additional comments

The paper is well written and does a good job characterizing many aspects of the fishery. This is a good first step towards solving this bycatch problem. As the author suggests further study is needed to specifically characterize the gill net fishery, which may identify some potential mitigation measures to be tested. I would suggest methods outlined in the following publication

Hall, M.A. (2009) First attempts to categorize and stratify nets for bycatch estimation, and for bycatch mitigation experiments. In: Proceedings of the Technical Workshop on Mitigating Sea Turtle Bycatch in Coastal Net Fisheries (ed. E. Gilman). IUCN, Western Pacific Regional Fishery Management Council, Southeast Asian Fisheries Development Center, Indian Ocean – South-East Asian Marine Turtle MoU, U.S. National Marine Fisheries Service, Southeast Fisheries Science Center, Gland, Switzerland; Honolulu, Bangkok, and Pascagoula, USA, pp. 42. ISBN: 1-934061-40-9.

·

Basic reporting

- The exposed information is confusing. It is recommended to organize all the text in a logical sequence.

- Some references are cited in the text, but not listed in the References section, and vice versa.

Experimental design

- The manuscript does not have an explicit objective. The authors should better detail it.

- Lack of details in the sampling design and regarding the Study Area. The authors should mention the different landscape formations of the main river and of the studied tributaries (later explored in the text).

Validity of the findings

- There are different values corresponding to the same data in the text and in its respective Tables. The order data are shown in the text is also different from that shown in the Tables. I highly recommend the authors to carefully review all the data presented in the text, comparing with the respective Tables, correcting when necessary, and standardizing the order data appear in both (text and Tables).

- Furthermore, data shown in the text and in the Tables should complement each other, not duplicate an information. The Tables should also be cited when their contents are mentioned in the text – some Tables (or part of them) were not cited in the manuscript.

- All the Tables have the same Title.

- The “n” values are incorrect.

- The number of decimals on the percentages are not standardized.

- There are some significant results with a “P” value higher than the defined Statistical significance.

- There are no Degrees of Freedom exposed, and some Statistical Tests do not have the test value.

- Some information appear in the wrong section.

- Some Results do not have a description of its methodology.

- Some points in Discussion are not mentioned in Results.

Additional comments

The manuscript is interesting and explores important issues on how artisanal fisheries affect the endangered Ganges River Dolphin population. These information can give insights into the dolphin conservation, being a valuable tool to management measures.
However, as a scientific study, it is important that the research question and the sampling design are clearly defined and described. Besides, the organization of all the information is essential to increase the reader’s interest and the visibility of the study. This can be improved in the present manuscript.
*There are some suggestions in the manuscript PDF.

---

## Round 0.2 · Minor Revisions

Please consider all the suggestions in the revised manuscript.

Reviewer 2 has stated that the current version has many grammatical errors and states that several recommendations that were suggested to improve the previous version were not followed.

Reviewer 1 ·

Basic reporting

There are errors in style and grammar.

Experimental design

No comments

Validity of the findings

No comments

Additional comments

This is the second submission of this article, focusing on Ganges River Dolphin bycatch in Nepal. The authors addressed previous comments and corrections thoroughly, and the resulting manuscript is clearer. They have also added the necessary supplementary information. There are however still some stylistic and grammatical errors:
Abstract: "Sixty-percent" should have a hyphen, for example.
Conclusions first sentence should be "The GRD is recognized as one of the most endangered cetaceanS in the world".
Needs one more thorough reading to make sure no small errors remain.

·

Basic reporting

This article fails to meet publication standards.

All sections need major revisions. There is too much to list. Please resubmit in a more finalized version

Experimental design

No comments

Validity of the findings

No comments

·

Basic reporting

Revise all the references, since the manuscript still has references not cited in the text.

Experimental design

The manuscript does not have an explicit objective. The objective mentioned in the last paragraph of the Introduction is a general description of the methodology, rather than the goal of the research.

Validity of the findings

No Comments

Additional comments

The manuscript is interesting and explores important issues on how artisanal fisheries affect the endangered Ganges River Dolphin population. This information can give insights into the dolphin conservation, being a valuable tool to management measures. However, as a scientific study, it is important that the research question is clearly defined.
An annotated PDF with general corrections is attached.

---

## Round 0.3 · accepted · Accept

Thank you for your edits. Your submission is Accepted.